# Sea-level records from the U.S. mid-Atlantic constrain Laurentide Ice Sheet extent during Marine Isotope Stage 3

T. Pico[1], J.R. Creveling[2] & J.X. Mitrovica[1]

The U.S. mid-Atlantic sea-level record is sensitive to the history of the Laurentide Ice Sheet as the coastline lies along the ice sheet's peripheral bulge. However, paleo sea-level markers on the present-day shoreline of Virginia and North Carolina dated to Marine Isotope Stage (MIS) 3, from 50 to 35 ka, are surprisingly high for this glacial interval, and remain unexplained by previous models of ice age adjustment or other local (for example, tectonic) effects. Here, we reconcile this sea-level record using a revised model of glacial isostatic adjustment characterized by a peak global mean sea level during MIS 3 of approximately $-40$ m, and far less ice volume within the eastern sector of the Laurentide Ice Sheet than traditional reconstructions for this interval. We conclude that the Laurentide Ice Sheet experienced a phase of very rapid growth in the 15 kyr leading into the Last Glacial Maximum, thus highlighting the potential of mid-field sea-level records to constrain areal extent of ice cover during glacial intervals with sparse geological observables.

[1] Harvard University, Department of Earth and Planetary Sciences, Cambridge, Massachusetts 02138, USA. [2] Oregon State University, College of Earth, Ocean, and Atmospheric Sciences, Corvallis, Oregon 97331, USA. Correspondence and requests for materials should be addressed to T.P. (email: tpico@g.harvard.edu).

Reconstructing the pace of ice growth towards the Last Glacial Maximum (LGM, 26 ka) is critical to our understanding of ice age climate and ice sheet stability. Nevertheless, global ice volume, or equivalent global mean sea level (GMSL), and the corresponding geographical distribution of ice remain uncertain through Marine Isotope Stage 3 (MIS 3; 60–26 ka) leading into the LGM[1,2]. Oxygen isotope records from marine sediment cores provide a proxy for global ice volume after correcting for temperature-dependent fractionation[3], however uncertainties in this correction and other complications in mapping isotope values to ice volumes have yielded estimates of peak MIS 3 GMSL that range from −30 to −60 m relative to present day[1]. Geological records of sea level during MIS 3 are sparse because ancient markers in the far field of former ice sheets are presently submerged, while those in the near field have been erased by the subsequent advance and retreat of the major continental ice sheets[4,5]. Moreover, glacial isostatic adjustment (GIA) and tectonic uplift contaminate the present-day elevation of available sea-level records[6,7]. Studies that applied GIA modelling to fit oxygen isotope records and geological sea-level markers have published discordant inferences of peak MIS 3 GMSL, varying from −85 m (ref. 8) to −55 m (ref. 9), and most recently −37.5 ± 7 m (ref. 10).

The geological markers of Pleistocene sea-level oscillations extending from Virginia to North Carolina in the Albemarle Embayment (Fig. 1), on the Laurentide Ice Sheet's (LIS) peripheral bulge, require a re-evaluation of ice volume and extent during MIS 3. This record indicates that MIS 3 relative sea level (RSL) reached present-day levels from ~50 to 35 ka in this region[11–16] (Fig. 1; Supplementary Table 1), but GIA calculations predict that these markers should presently be found as much as ~70 m below sea level[8]. Tectonic uplift of the markers is insufficient to explain their present-day elevation[17,18] and sediment compaction has led to only minor subsidence in this region[19].

Here, we present a new set of GIA calculations that explore the sensitivity of the predictions to peak GMSL and LIS geometry during MIS 3. We conclude that a revised GIA model can reconcile the MIS 3 sea-level record at the Albemarle Embayment under two conditions: (1) peak GMSL reached near −40 m and (2) the eastern sector of LIS was significantly reduced during MIS 3 compared with previous reconstructions of ice extent.

## Results

**The U.S. Mid-Atlantic sea-level record.** The Albemarle Embayment geological record includes interfluvial, estuarine, intertidal and shallow marine lithofacies arranged in depositional sequences that record repeated sea-level highstands dated primarily by optically stimulated luminescence to MIS 5e, 5c, 5a and 3 (ref. 11) (Fig. 1; Supplementary Table 1; Supplementary Note 1). We adopt the minimum elevation of terrestrial facies and the maximum elevation of marine facies as upper and lower bounds, respectively, of MIS 5a (~80 ka) and mid-MIS 3 (50–35 ka) sea level. For the MIS 5a data (Fig. 1; Supplementary Table 1), we bound a cluster of sea-level data from 2.5 to 7 m in agreement with previous assessments of sea-level records in the region[20]. We assume that rare terrestrial markers found at elevations below this range do not represent a constraint on the MIS 5a highstand, but rather a lower sea level reached during late MIS 5a or MIS 4. Furthermore, calculations described below (and detailed in Supplementary Note 3) demonstrate that RSL predictions for MIS 3 are relatively insensitive to the height of sea level during MIS 5a.

For the MIS 3 interval spanning 50–35 ka, three marine indicators constrain RSL to be above −0.9, −3 and −2 m (ref. 11). We thus adopt the elevation of the highest of these

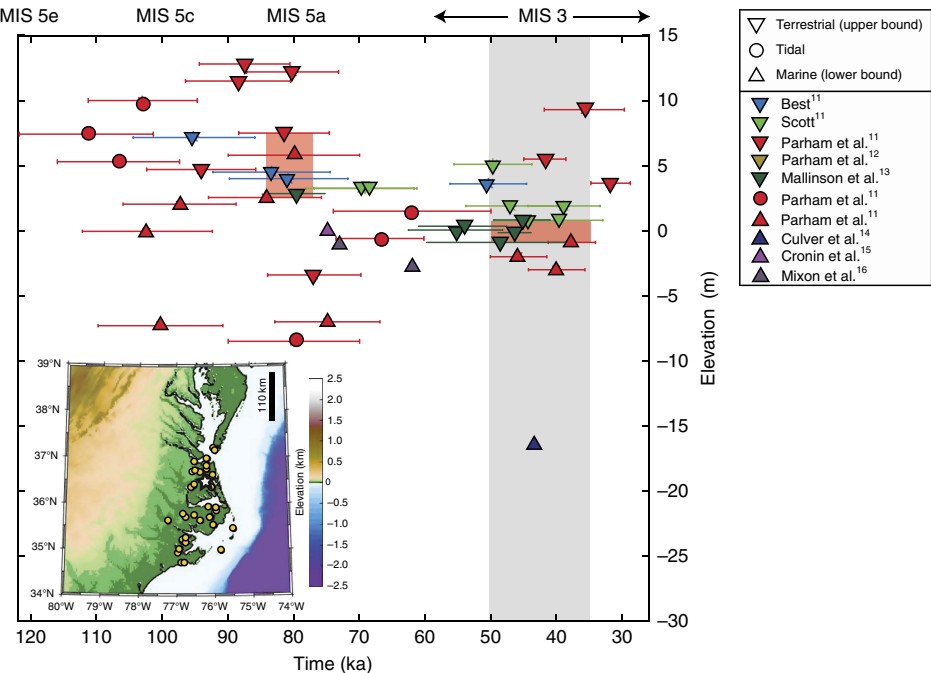

**Figure 1 | Present elevation of sea-level indicators from the Last Interglacial to the Last Glacial Maximum for the Albemarle Embayment.** Field localities are shown by yellow dots on the inset map. Upwards pointing triangles represent marine indicators (lower bound), downwards-oriented triangles represent terrestrial indicators (upper bound), and circles designate tidal facies. Error bars span 2-σ age uncertainties on individual sea-level data. Marine Isotope Stages 5e, 5c, 5a and 3 are labelled at 120, 100, 80 and 60–26 ka, respectively. The shaded region covers the time interval examined within the present analysis and the orange rectangles mark the bounds on MIS 5a and MIS 3 sea level based on the plotted data (MIS 5a: 2.5–7.5 m; MIS 3: −1 to 1 m). The white star on the inset map marks the location of RSL predictions presented herein.

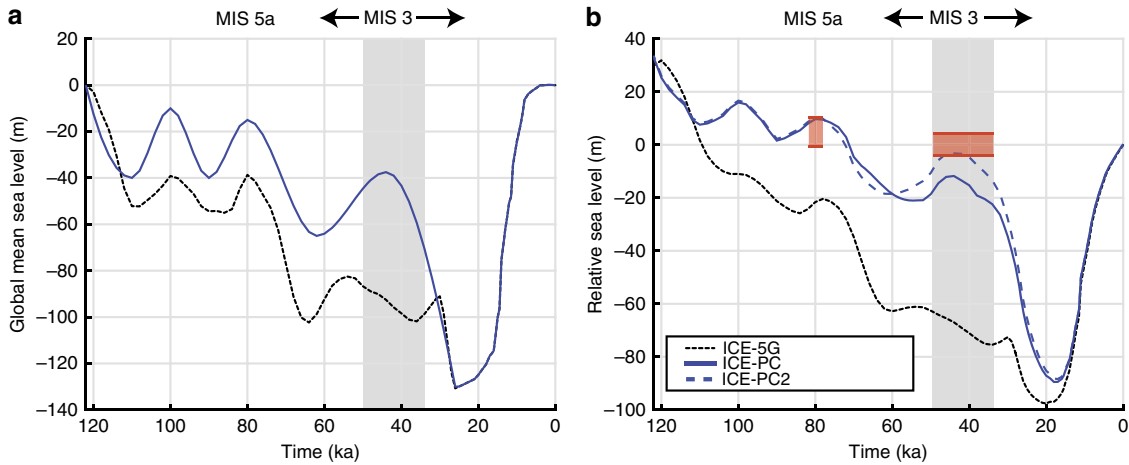

**Figure 2 | Global mean sea-level curves and relative sea-level prediction.** (**a**) Global mean sea-level curve for Version 1.2 of ICE-5G (dotted black line) and the ICE$_{PC}$ (blue line) ice histories. The GMSL curve for the ICE$_{PC2}$ history is identical to the curve for ICE$_{PC}$. (**b**) Relative sea-level predictions for the reference site in the Albemarle Embayment based on the ICE-5G (dotted black), ICE$_{PC}$ (solid blue) and ICE$_{PC2}$ (dashed blue; ice-free eastern sector of the Laurentide Ice Sheet from 80 to 44 ka) ice histories. Orange rectangles span the observational constraints on peak MIS 5a and 3 sea level including the $\pm 3$ m paleotidal uncertainty (see Fig. 1). For MIS 5a, this range is $-0.5$ to 10.5 m, and at 44 ka during MIS 3 the range is $-4$ to 4 m. The grey-shaded region spans the MIS 3 time interval examined within the present analysis.

marine indicators, $-0.9$ m, as the lower bound. Regarding the upper bound, three terrestrial indicators, with ages between 50 and 35 ka, show a consistent constraint on the sea-level highstand of 1 m. Two terrestrial indicators dated to earlier in this time window[13] are found at lower elevations, however these may represent deposition during a time of rising sea level rather than during the peak sea-level highstand. We adopt the terrestrial indicators at 1 m as the upper bound on sea level, yielding a range of $-1$ to 1 m. We apply an elevation error of $\pm 3$ m that reflects reconstructed paleo-tidal range for the region that may have been up to three times greater than the present amplitude of $\sim 1$ m (ref. 21). The geological sea-level constraints we adopt below (for example, Fig. 2) incorporate these broader uncertainties.

**Models of glacio-isostatic adjustment.** Ice sheet growth and melt produces a complex spatio-temporal pattern of sea-level change[22]. To predict the present elevation of sea-level markers, we perform calculations based on the sea-level theory and pseudo-spectral algorithm described by Kendall et al.[23] with a spherical harmonic truncation at degree and order 256. The calculations include the impact of rotation changes on sea level[24], evolving shorelines and the migration of grounded, marine-based ice[23,25–27]. We report RSL predictions at a representative site within the Albemarle Embayment (white star on inset of Fig. 1) for a representative time (44 ka) within the middle of MIS 3 (50–35 ka). This representative site lies within the latitudinal range of the reported geological sea-level markers used to define the bound on local peak MIS 3 sea level (Fig. 1). We have found that RSL highstand predictions for this reference site differ from field locations by less than 0.5 m. We deem simulations acceptable if they satisfy the aforementioned bounds for both MIS 5a and MIS 3 (Fig. 2b, orange rectangles).

Our numerical predictions require models for Earth's viscoelastic structure and the history of global ice cover. We begin by adopting an Earth model with upper and lower mantle viscosities of $0.5 \times 10^{21}$ Pa s and $1.5 \times 10^{22}$ Pa s, respectively; this radial profile is consistent with inferences based on globally distributed ice age data sets[28] and geological data along the U.S. mid-Atlantic[29,30]. Our initial GIA calculation adopts Version 1.2 of the ICE-5G ice history, characterized by a GMSL

fall from $-87$ to $-100$ m throughout MIS 3 (ref. 8) (Fig. 2a; dotted black line); in this calculation, we make the standard assumption that, for any pre-LGM time step, the geometry of global ice cover was identical to the post-LGM ice distribution with the same GMSL value[31]. We explore alternatives to the GMSL history, ice geometry and viscosity profile in the discussion below. Using the combination of the ICE-5G model and Earth structure described above, we predict mid-MIS 3 sea level (at 44 ka) at the Albemarle Embayment reference site to be –67 m (Fig. 2b; dotted black line), grossly misfitting (by $\sim 70$ m) the observational constraints (Fig. 1). The misfit is $\sim 25$ m for the MIS 5a record (Fig. 2b). The level of misfit to the MIS 3 record highlights the enigmatic nature of the sea-level record in Fig. 1 and motivates the present study.

Many previous inferences of GMSL during the last glacial phase, particularly MIS 3, reconstruct higher peak sea level (smaller global ice volume) than adopted in the ICE-5G history[9,32,33]. To proceed in our analysis, we revise the ICE-5G ice history on the basis of results from two recent GIA analyses. First, following the Pico et al.[10] analysis of sediment core records from the Bohai Sea, peak GMSL during MIS 3 is placed at $-37.5$ m at 44 ka. Second, we adopt GMSL values of $-15$ and $-10$ m for MIS 5a and 5c, respectively; these values are within bounds (5a: $-18$ to 0 m, 5c: $-20$ to 1 m) derived by Creveling et al.[30] on the basis of globally distributed sea-level markers from both periods. The GMSL curve for the revised ice model, ICE$_{PC}$, is shown in Fig. 2a (blue line). The RSL prediction based on this model (Fig. 2b, solid blue) maintains the assumption that the pre-LGM global ice geometry is equivalent to the deglacial phase whenever GMSL values are equal. This prediction is consistent with observational constraints for the MIS 5a highstand, but misfits MIS 3 data. Notably, this Earth–ice model pairing predicts a peak RSL of $-12$ m at 44 ka, well below the observational bounds of $-4$ to 4 m.

We performed a suite of simulations to explore the sensitivity of our predictions to the adopted ice history. Specifically, we generated 100 synthetic ice histories in which we varied GMSL randomly across the glacial phase but confined GMSL to $-37.5 \pm 7$ m at 44 ka (ref. 10), and to $-15$ m and $-10$ m, for MIS 5a and MIS 5c, respectively (Fig. 3a, blue lines; see Methods for detailed ice history construction). Using these ice histories, we

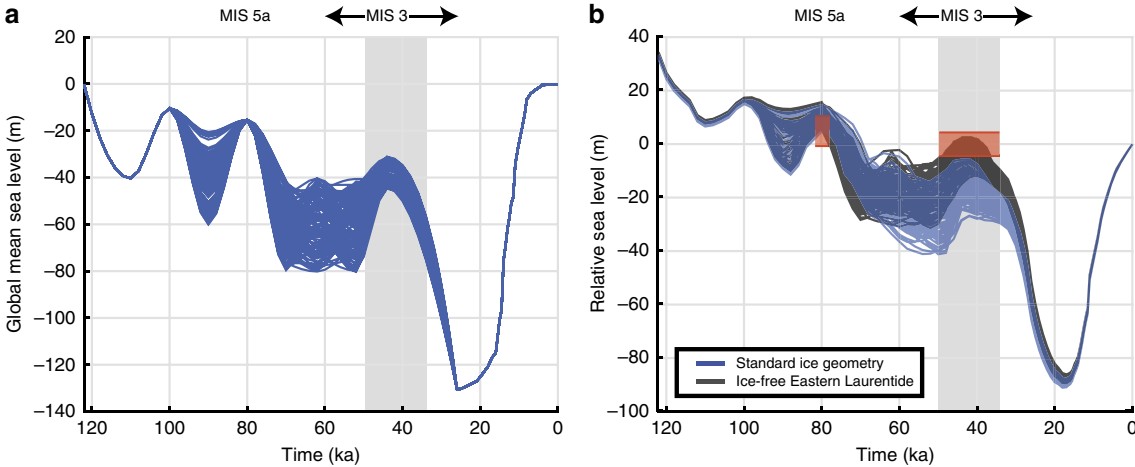

**Figure 3 | The effect of varying ice history on relative sea-level prediction.** (**a**) Global mean sea level curves for 100 randomly generated ice histories that pass through −37.5 ± 7 m at 44 ka (blue lines), −15 m at MIS 5a and −10 m at MIS 5c. (**b**) Relative sea level predictions for the Albemarle Embayment reference site based on the ice histories shown in frame (**a**). The results of calculations that assume identical pre-LGM and post-LGM ice geometries when global mean sea level values are the same are plotted as blue lines; the calculations that assume an ice-free eastern Laurentide from 80 to 44 ka are shown as black lines. Orange rectangles span the adopted observational constraints on peak MIS 5a and MIS 3 sea level including the ±3 m paleotidal uncertainty. The grey-shaded region spans the MIS 3 time interval examined within the present analysis.

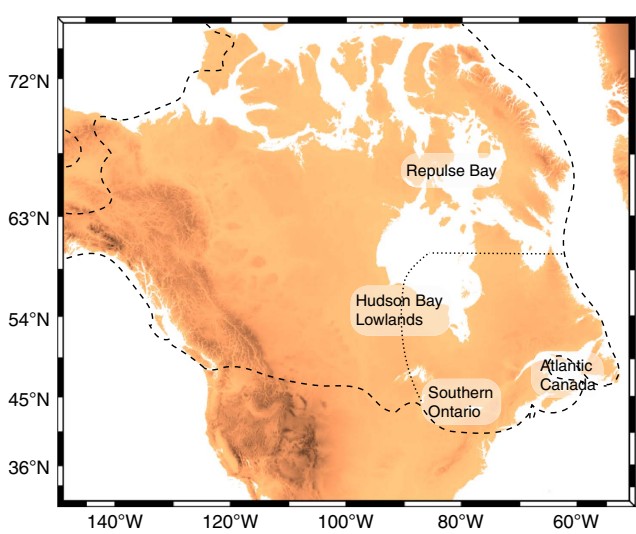

**Figure 4 | Geographic map showing the extent of the Laurentide Ice Sheet.** At LGM in the ICE-5G model (dashed black line) and showing the eastern extent of the ice model ICE$_{PC2}$ from MIS 5a to MIS 3 (dotted black line). Repulse Bay, Hudson Bay Lowlands, Southern Ontario, and Atlantic Canada are all sites that have been reported as deglaciated at MIS 3 in Dalton et al.[34]

predicted RSL at the reference site within the Albemarle Embayment using the standard treatment for the pre-LGM ice distribution; that is, this distribution matches the post-LGM geometry when the GMSL values are the same (similar to ICE$_{PC}$). In this case, the predicted RSL ranges from −26.5 to −7.5 m at 44 ka (blue lines, Fig. 3b), and thus all 100 simulations predict a peak RSL that falls outside the observational constraints.

**Revising the geometry of the Laurentide Ice Sheet during MIS 3.** We next explored the impact of changing the geometry of the LIS on the local RSL predictions at the Albemarle Embayment. While few field data constrain the evolution of the LIS before the LGM[2,4,5], a recent field-based study by Dalton et al.[34] suggests that

large portions of eastern Laurentia were ice-free during MIS 3 (Fig. 4). This conclusion implies limited or no ice growth from MIS 5a to MIS 3 within large areas of the sector of the LIS closest to the Albemarle Embayment. To investigate the effect of this revised ice geometry on RSL predictions, we constructed an ice model, ICE$_{PC2}$, with a GMSL history identical to that shown by the blue line in Fig. 2a, but that was distinguished from the ICE$_{PC}$ history in the following ways: (1) the eastern sector of the LIS is ice-free from 80 to 44 ka, consistent with the conclusions of Dalton et al.[34]; and (2) the ice removed in this exercise, equivalent to 6.8 m of GMSL, is distributed uniformly over the western sector of the LIS, and the Cordilleran and Fennoscandian Ice Sheets. The latter resulted in an increase in ice thickness of ∼170 m in each region (Fig. 4; See Methods for details on ice model construction). The post-LGM ice geometry remains identical to the ICE-5G and ICE$_{PC}$ models, and thus we no longer assume that global ice geometry prior and subsequent to the LGM were identical whenever the GMSL values match. The simulation, based on this revised ICE$_{PC2}$ ice model and the viscoelastic Earth model discussed above, predicts a RSL of −3 m at 44 ka, consistent with the sea-level record at the Albemarle Embayment (Fig. 2b).

## Discussion
What physics underlies this improved fit to the MIS 3 sea-level record in the mid-Atlantic coastal region? Crustal deformation and the direct gravitational effect of the surface load dominate sea-level predictions in this location and yield a RSL history that departs significantly from GMSL. To assess the relative contribution of each process, we decomposed the RSL prediction based on ice models ICE$_{PC}$ and ICE$_{PC2}$ into these two components (Supplementary Fig. 1; Supplementary Note 2). Ice model ICE$_{PC2}$ is defined by a reduced eastern sector of the LIS, and therefore a smaller surface load compared with the ICE$_{PC}$ model. This smaller ice load results in a reduced direct gravitational effect (expressed as a sea-level fall) and a reduced crustal deformation (a smaller upward deflection of the Earth's surface, expressed as sea-level rise), compared to the ICE$_{PC}$ prediction. The latter effect dominates, resulting in a net sea-level rise compared to the ICE$_{PC}$ ice distribution and the fit evident in Fig. 2b (dashed blue line). We explore the sensitivity of the RSL decomposition to variations in Earth structure in Supple-

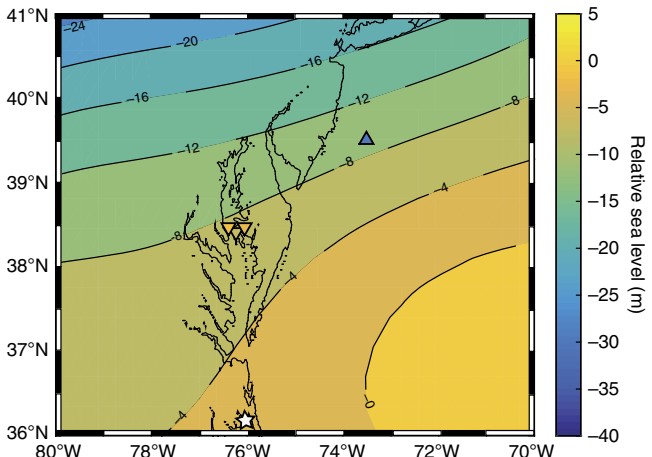

**Figure 5 | Map of predicted relative sea-level based on ice history ICE_PC2 at 44 ka.** The Albemarle Embayment site is shown by the white star. Elevation of sea level data are plotted as triangles at the Chesapeake Bay (−7.6 m, −1.7 m and −3.4 m) and the Hudson shelf (−30 m), where the colour represents the elevation shown by the colour bar. Upper bounds on sea level are represented by downwards pointing triangles, while lower bounds are plotted as upwards pointing triangles.

mentary Note 4, where we adopt the ICE_PC2 ice history and consider a range of lithospheric thicknesses, and upper and lower mantle viscosities (see Supplementary Fig. 2).

We performed several additional sensitivity tests related to the ICE_PC2 simulation. For example, we once again randomly generated 100 simulations, and constructed ice histories with the GMSL curves shown in Fig. 3a, but now assumed (as in ICE_PC2) that the eastern sector of the LIS was ice-free from 80 to 44 ka. In these simulations, the predicted peak RSL at the reference site in the Albemarle Embayment ranged from −11 to −1 m at 44 ka (black lines, Fig. 3b). Nearly half (46 of 100) of these simulations yielded predictions consistent with observational constraints on the MIS 3 highstand. We also constructed ice histories in which we varied peak GMSL during MIS 5a and MIS 5c in the range of −16 to 0 m and −20 to 0 m, respectively, within bounds derived by Creveling et al.[30], to test the sensitivity of the RSL predictions to this level of uncertainty. This suite of simulations perturbed the peak RSL prediction during MIS 3 by less than 0.7 m, and thus our conclusions in regard to ice cover during MIS 3 are insensitive to the GMSL values adopted for MIS 5a and 5c.

We also performed tests in which the geometry of ice removed from the eastern sector of the LIS was varied by shifting the upper latitudinal limit of the ice-free region, and by considering scenarios in which various regions within the eastern sector of the LIS remained glaciated (see Supplementary Fig. 3). We conclude that increasing the ice in the eastern sector by 20% of the ice volume removed to construct the ice history ICE_PC2 (or 1.5 m GMSL) from 80 to 44 ka, and shifting the distribution of this ice within the sector, can lower the RSL predictions by up to 3 m (Supplementary Table 2). Finally, to assess the sensitivity of the predictions to the adopted Earth model, we ran simulations in which the lower mantle viscosity was both increased and decreased by $5 \times 10^{21}$ Pa s. The predicted RSL at 44 ka was perturbed by a maximum of 3 m at the Albemarle Embayment (Supplementary Fig. 4). We also ran simulations with an additional six Earth models to assess the sensitivity of RSL predictions to lithospheric thickness and upper mantle viscosity (details in Supplementary Note 4; Supplementary Fig. 5).

Finally, additional constraints exist on RSL during MIS 3 at other sites along the U.S. East Coast. For example, a marine indicator at −30 m on the Hudson Shelf of age ∼45 ka provides

a lower bound on sea level at this site[35], while terrestrial facies in the Chesapeake Bay indicate that local sea level was below approximately −7.6 m sometime during the interval 50–35 ka (ref. 36). RSL predictions based on ice model ICE_PC2 agree well with both these constraints (Fig. 5).

Sea-level records from the mid-Atlantic coastal plain show that MIS 3 sea level reached present-day levels. These observations have been considered enigmatic given that previously published GIA models predict RSL in this region to be more than 60 m below present level at MIS 3 (ref. 18). However, we have shown that GIA models can be reconciled with the observational record from the Albemarle Embayment under two conditions: (1) global mean sea level during MIS 3 reached approximately −40 m, consistent with a recent analysis of sediment core records from the Bohai Sea[10] and within the bounds of several earlier studies[37–43]; and (2) the eastern sector of the LIS remained largely ice-free over an extended period from MIS 5a through mid-MIS 3, consistent with recent field-based evidence[34]. Rigorous tests of our conclusions regarding ice extent will require improved chronological control on geological indicators of LIS extent, as presented in, for example, Clark et al.[5], Curry et al.[44], Colgan et al.[45] and Dalton et al.[34]. Our inference, if robust, implies that the LIS rapidly advanced during the ∼15 kyr from mid-MIS 3 to the Last Glacial Maximum. Moreover, our study highlights the potential of mid-field sea-level records to constrain ice load locations over glacial intervals where geologic evidence of ice cover is poorly preserved.

## Methods

**Ice history construction.** We created 100 ice models that are distinguished from Version 1.2 of ICE-5G (Peltier & Fairbanks, 2006) by their GMSL history before the Last Glacial Maximum. These ice histories randomly sample GMSL values under the following constraints: at LIG, 122 ka, GMSL = 0; at 110 ka, MIS 5d, GMSL = −40 m; at MIS 5c, GMSL = −10 m; at MIS 5a, GMSL = −15 m; during MIS 3 before 44 ka, GMSL varies in the range −40 to −80 m; at 44 ka, the GMSL value is pinned within the range −37.5 ± 7 m (Fig. 3a). We also impose the constraint that sea level must fall between 44 ka and LGM. We produce two sets of 100 ice models from these GMSL curves. The first set is constructed by assuming that ice geometry in the pre-LGM period is identical to the time in the post-LGM period with the same GMSL value. The second set assumes that a large portion of the eastern sector of the LIS is ice-free from 80 to 44 ka (see Fig. 4). In the latter case, the ice volume removed from the eastern sector of the LIS (equivalent GMSL of 6.8 m) is distributed over the LGM extent of the western sector of the LIS, the Cordilleran Ice Sheet and over Fennoscandia, resulting in an increase in ice thickness of 170 m in these regions.

We note that while RSL predictions at the Albemarle Embayment are sensitive to ice geometry of the eastern LIS, these predictions will be relatively insensitive to the geographic distribution of ice cover in regions in the far field of the mid-Atlantic U.S. East Coast. As an example, we performed a simple test in which the ice removed from the eastern sector of LIS to construct the ICE_PC2 model was distributed uniformly over Fennoscandia and Antarctica (rather than over the western sector of the LIS) and found that the predicted sea-level highstand at our test site at 44 ka was only perturbed by 0.6 m.

**Data availability.** The data sets generated during and/or analysed during the current study are available from the corresponding author on request.

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

## Acknowledgements

T.P. was supported by the NSF-GRFP and the Harvard University. J.R.C. acknowledges a donation from the G. Unger Vetlesen Foundation to Oregon State University. J.X.M. was funded by the Harvard University.

## Author contributions

T.P. conducted the simulations and analysed the results. J.R.C., J.X.M. and T.P. conceived the project. All authors contributed extensively to this work. T.P. and J.X.M. wrote the manuscript.

## Additional information

**Competing interests:** The authors declare no competing financial interests.

