## [Peer Review File · Nature Communications]

Reviewers' comments:

Reviewer #1 (Remarks to the Author):

Review of 'Sea Level Records from the U.S. Mid-Atlantic Constrain Laurentide Ice Sheet Extent During Marine Isotope Stage 3' by Pico et al.

This manuscript describes a highly novel study in which sea-level indicators from the US east coast are used to constrain not only global mean sea level during Marine Isotope Stage (MIS) 3, but also the configuration of the Laurentide Ice Sheet at this time, using a Glacial Isostatic Adjustment modelling approach. The manuscript is very well written, the methodology is sound, and the results are novel. There are a few areas where I feel that additional caveats or sensitivity studies are required to ensure that the results are robust, but these issues should be relatively easy to address.

Main comments

1. The decision to adopt a range of +2.5 m to +7 m for local sea levels during MIS 5a is justified in the main text by reference to the existence of paired marine-terrestrial indicators from this period (line 68). However, examination of Figure 1 reveals that if sea level had been in the upper part of your chosen range then three terrestrial indicators from this period would have been submerged. Similarly, if sea level had been closer to the lower bound then one of the marine constraints would have been located above sea level. Similar arguments could be made about the range adopted for the MIS 3 period. I think that you justify the choice of range by noting that some of the indicators are not well dated, but this information is only really found by studying the supplementary information; the sentence about dating uncertainty in the main text (line 71) is not very clear. The overall conclusions of the article are not affected by these details since your main result relies on determining a scenario that can give high enough relative sea-level (RSL) values to explain the marine indicators, but it would be useful if you could explain a little more clearly in the main text why some of the sea-level indicators shown in Figure 1 seem to be contravened by your choice of RSL range.

2. In this study, RSL predictions are generated for a single location, indicated by the white star in Figure 1. In reality, RSL change will not have been the same over the whole area covered by the field data (please add a scale to the inset map in Figure 1). Therefore, in the same way that you explore the sensitivity of the model output to ice history and earth rheology, please also provide brief justification that the RSL predictions generated for the location of the white star are representative of RSL change across the whole study area. In particular, please confirm that the predictions sufficiently represent RSL change at the location of the key sea-level indicators (those used to define the RSL range for MIS 5a and MIS 3).

3. Why do you explore the sensitivity to lower mantle viscosity but not upper mantle viscosity? I note that a recent study by Love et al. [2016], which uses US East Coast RSL data to infer regional mantle viscosity values, determines an optimal upper mantle viscosity value for this region that is greater than the value used in this study. The sensitivity of the RSL predictions to upper mantle viscosity should be briefly explored, in the same way that it is for lower mantle viscosity.

Love, R., G. A. Milne, L. Tarasov, S. E. Engelhart, M. P. Hijma, K. Latychev, B. P. Horton, and T. E. Tornqvist (2016), The contribution of glacial isostatic adjustment to projections of sea-level change along the Atlantic and Gulf coasts of North America, *Earths Future*, 4(10), 440-464, doi:10.1002/2016ef000363.

4. Related to the above point: you discuss (on line 208) the trade-off between a reduction in the direct gravitational effect (which results in sea-level fall in the scenario considered here) and a reduction in the crustal deformation component (which results in sea-level rise). Will the

magnitude of the crustal deformation component depend on upper mantle viscosity, and could the adoption of a different upper mantle viscosity lead to a scenario in which the crustal deformation effect does not dominate (see line 208)? The reduction in the magnitude of the peripheral bulge seems to be an important factor that enables you to fit the sea-level data (dashed blue line, Figure 2B), so uncertainty on the magnitude of this effect (due to uncertainty in lithosphere thickness or upper mantle viscosity) should be explored.

5. The idea of using global ice volume to define the ice extent at a given time (lines 114-115) is a neat way of getting around the problem of working out how ice was spatially distributed during the build-up to the LGM. However, as demonstrated by your results relating to ice extent in the eastern Laurentide, this assumption will not always have held in reality. Given that the partitioning of ice mass between the different ice sheets has a fundamental influence on the pattern of RSL at a given time; would your results still hold if the distribution of ice outside the Laurentide Ice Sheet (LIS) was very different to that assumed here? Addressing this issue in a robust manner is outside the scope of this study, but a brief discussion of whether it could negate the results is warranted.

6. Changing the geometry of ice in the eastern sector of the LIS seems to have a significant effect on RSL predictions at the study site (e.g. lines 228-231) so I took some time to try to understand how the various ice history scenarios differed. However, I found it difficult to follow the description of the different ice sheet geometries (lines 88-99 in the supplementary information), partly because the text did not always seem to agree with the information shown in Supplementary Figure 2. For example, panel B of this figure shows configuration PC2, in which the latitudinal boundary for no-ice is apparently 60° N (line 91 of supplementary material). But in the figure, the southern margin of the ice sheet does not just terminate at 60° N across eastern Laurentide; do you also invoke some (sensible) assumption relating to the fact that ice extent will also be governed by the underlying topography? What are the implications of this for PC3 and PC4 (which are not shown, but apparently have a different latitudinal extent)? I was also confused by the statement that Baffin Island is not glaciated in Geometry 2 and Geometry 3 (lines 94-95), since it does appear to be partly glaciated in Supplementary Figure 2D. Does the statement just relate to the southern part of the island? There is also reference to Figure 4B in the caption to Supplementary Figure 2, which does not seem to exist.

Minor comments

1. Title: Should 'Sea-Level Records' be hyphenated (you seem to follow this rule in all other cases where 'sea level' is used as a compound adjective)?

2. Line 13-14: 'paleo sea-level values near present-day' reads a little odd as 'present-day' refers to a time; perhaps rephrase to clarify that you are referring to the fact that MIS 3 sea level indicators are found close to the elevation of the current shoreline.

3. Lines 52-53: the text on these lines is a little ambiguous: it is not clear whether you are suggesting that previous GIA calculations predict that the land where the shoreline records are found was 70 m below sea level during MIS 3, or whether previous GIA calculations predict that the MIS3 shoreline records should currently be 70 m below sea level, rather than close to present sea level. (I think the latter is correct)

4. Please briefly mention whether sediment compaction is likely to have affected any of the sea-level indicators.

5. Lines 13-114: '...we make the standard assumption...' – is this just an assumption that this adopted throughout this study, or is it an assumption that has been used in previous studies? If the latter, please include a reference.

6. Line 117: it is not quite clear what 'this combination' refers to, because the train of logic is

slightly interrupted by the mention of exploring other model combinations in the previous sentence.

7. Lines 182-184: In the description of the scenarios where the eastern sector of the LIS is ice free you mention that ice is re-distributed over the western LIS and the Fennoscandian Ice Sheet (FIS). Please give an indication of what this means in terms of ice thickness added to the western LIS and the FIS, so that readers can determine whether this is a reasonable modification to the existing ice history in these regions.

8. Line 219: the ranges quoted seem to be incorrect (correct values are given in the supplementary material)

9. Figure 3B: It is difficult to distinguish the blue lines from the purple lines

10. Since you use a continuous colour scale in Figure 5, please label the contours to make it easier to interpret this figure

11. Supplementary Figure 1: please clarify in the caption to this figure that predictions are for the location of the white star in Figure 1

Pippa Whitehouse

Reviewer #2 (Remarks to the Author):

This manuscript examines some previously enigmatic MIS 3 sea level constraints along the U.S. East Coast from a GIA modeling perspective. They show that the fairly "high" elevations of the relative sea level markers can be explained using a revised geographic extent of ice cover in the Laurentide ice sheet during this time interval. This paper is a valuable contribution and is appropriate for the audience of Nature Communications both in terms of the value of this particular study in explaining these MIS 3 data (that I have personally wondered about) and also underscores the ever important message to the broader community that local, or relative sea level can indeed be very different than global mean sea level, which remains an extremely valuable message for the broader community to appreciate.

My comments are minor, and I would anticipate that this manuscript could be accepted after some of the minor revisions below are addressed:

Line 35: I think you mean "reconstructions" rather than "complications"

Line 38: instead of "ice cover" I think it is clearer to say "former ice sheets" or "former ice sheet extent"

Line 41: consider re-phrasing to "present-day elevation"

Line 61: spelling of estuarine

Line 96: theory of what?

Line 100-102: What is the effect of this assumption? Ie., how valid is it to assume that sea level is the same at the site of the white star and for the remaining sites where the data come from? Is there significant regional heterogeneity in the RSL signal?

Figure 2. The subscripts of the different model names is near impossible to read with the naked eye. The black dotted line seems a lot thicker in the legend for some reason.

Line 137: what is "this ice history" ICE-5G? If yes, then explicitly say so.

Figure 3. The colors (in particular the purple and especially when overlaid by the orange) are very hard to discern here. Perhaps making the purple lines black or gray would increase the ability to distinguish this group of curves visually.

Figure 4. It is not apparent from the one dotted line, which side of the dotted line the ice is sitting on (though the text explains it). Perhaps a white transparent overlay for the area of the MIS 5a to 3 ice sheet would help here. Perhaps also saying that the dotted line shows the eastern extent instead of extent, would also help?

Line 252: insert the word "that" after "show"

Supplementary Fig. 2: you reference figure 4B multiple times here, but there is no panel B in Fig. 4 of the manuscript. Upon examining this figure, I am now very confused by figure 4 in the text. There seems to be a discrepancy here. In caption for this supplementary figure you need to say what time frame is represented in these diagrams... i.e., geography distribution of ice WHEN?

Reviewer #3 (Remarks to the Author):

This manuscript presents a new comparison of existing sea level data from the U.S. mid-Atlantic with an ensemble of Glacial Isostatic Adjustment (GIA) model simulations for MIS3. Recently published data (Pico et al., 2016, QSR) suggest that global mean sea level was higher than previous estimates, implicating a reduced global ice volume during this time. While considering a variety of parameters in the GIA, the authors indicate that US Mid-Atlantic sea level records for MIS3 can only be explained with a reduced ice volume over the eastern sector of the Laurentide Ice Sheet (LIS). The authors appear to have done their due diligence in assessing the parametric uncertainty in their GIA model (mantle viscosity, MIS5a&MIS5c sea level uncertainty, eastern LIS volume reconstruction uncertainty, etc). The findings because most pre-LGM ice extent and volume constraints are poor due to the destruction of potential records by the LIS at maximum extent. This work will provide useful constraints on ice sheet boundary conditions for the modelling community.

The manuscript is well written and provides a clear description of the mechanisms controlling U.S. mid-Atlantic sea level change during MIS3. I support this manuscript for publication, but I have a few questions and minor comments that should be considered to clarify some of the methods used in this analysis:

Line 74: It is not completely clear to me why the authors select 1m as the upper bound for their MIS3 sea level range. The authors use Qtp-1 as their upper terrestrial limiting sample, but there appear to be other terrestrial records from Mallinson et al. (2008) that are a bit lower. It does not appear it will change the conclusions, but I would like a better description as to why Qtp-1 was selected for the upper bound.

Line 140-141: Please report the full range of GMSL values for MIS 5a and 5c from Creveling et al and why you select -15m and -10m, respectively, as the optimal values for your initial model runs. This gets discussed later in the sensitivity tests (lines 211-223), but it would be good to introduce

the full range here.

Figure 2b and 3b: Do the MIS 5a and MIS 3 sea level boxes include the +/- 3m paleotidal uncertainty. If so, perhaps consider mentioning this in the figure description so there is not confusion as to why vertical range is larger than in figure 1.

Figure 3b: Would it be possible to increase the transparency of the blue 'standard ice geometry lines.' As is, it is impossible to see the bottom of the range of the purple lines ('ice-free Eastern Laurentide').

Line 215-216: "-11 m to -1 m": check the negative sign in front of -1. It looks like it is a dash.

Line 219: The Creveling et al. ranges, "0-16 m" and "0-20 m", should be negative, right?

Supplementary Figure 1: flip x-axis so that it follows the same direction as the figures of the main text.

Supplemental Line 78: Please provide further description as to how you distribute the eastern sector LIS volume over western LIS and Fennoscandia. Is this equally distributed? Are there any other RSL records that could constrain where you put this ice? Why not put this ice in Antarctica?

Response to the Reviews

We thank the three referees for their very positive and constructive reviews of our original manuscript. Our appreciation is reflected in the acknowledgements. As we detail below, we have revised the manuscript to address each of the comments raised by the reviewers. In the following, we intersperse the reviewers' comments (black font) with our responses (in blue font).

Response to Reviewer #1:

Review of 'Sea Level Records from the U.S. Mid-Atlantic Constrain Laurentide Ice Sheet Extent During Marine Isotope Stage 3' by Pico et al.

This manuscript describes a highly novel study in which sea-level indicators from the US east coast are used to constrain not only global mean sea level during Marine Isotope Stage (MIS) 3, but also the configuration of the Laurentide Ice Sheet at this time, using a Glacial Isostatic Adjustment modelling approach. The manuscript is very well written, the methodology is sound, and the results are novel. There are a few areas where I feel that additional caveats or sensitivity studies are required to ensure that the results are robust, but these issues should be relatively easy to address.

Main comments

1. The decision to adopt a range of +2.5 m to +7 m for local sea levels during MIS 5a is justified in the main text by reference to the existence of paired marine-terrestrial indicators from this period (line 68). However, examination of Figure 1 reveals that if sea level had been in the upper part of your chosen range then three terrestrial indicators from this period would have been submerged. Similarly, if sea level had been closer to the lower bound then one of the marine constraints would have been located above sea level. Similar arguments could be made about the range adopted for the MIS 3 period. I think that you justify the choice of range by noting that some of the indicators are not well dated, but this information is only really found by studying the supplementary information; the sentence about dating uncertainty in the main text (line 71) is not very clear. The overall conclusions of the article are not affected by these details since your main result relies on determining a scenario that can give high enough relative sea-level (RSL) values to explain the marine indicators, but it would be useful if you could explain a little more clearly in the main text why some of the sea-level indicators shown in Figure 1 seem to be contravened by your choice of RSL range.

The reviewer raises several important issues in regard to our stated bounds on local RSL during MIS5a and MIS 3 within the Albemarle Embayment. We respond to each of these issues in turn.

Consider MIS 5a. Sea-level markers dated to MIS 5a (shown in Figure 1) include a number of contradictory constraints in which marine indicators lie above terrestrial data. However age uncertainties for the MIS 5a period are generally large (~6-10 m) compared with age uncertainties during MIS 3 (~3-5 m). Therefore, we selected a bound on RSL that was informed by this larger uncertainty, and that was consistent with previous studies of sea level in this region (i.e. Potter & Lambeck 2003, Wehmiller et al. 2004). However, in introducing the bound on MIS 5a sea level in the manuscript we should have emphasized that the choice has little impact on predictions of RSL at MIS 3; this insensitivity was demonstrated by the analysis in original manuscript in which GMSL values were varied across MIS 5a and MIS 5c (Supplementary Material lines 115-119). We have revised the text dealing with bounds on MIS 5a sea level to read (lines 68-75):

"For the MIS 5a data (Figure 1; Table S1), we bound a cluster of sea-level data from 2.5 – 7 m (Parham et al., 2013), in agreement with previous assessments of sea level records in the region (Wehmiller et al., 2004). We assume that rare terrestrial markers found at elevations below this range do not represent a constraint on the MIS 5a highstand, but rather a lower sea level reached during late MIS 5a or MIS 4. Furthermore, calculations described below (and detailed in the Supplementary Material) demonstrate that RSL predictions for MIS 3 are relatively insensitive to the height of sea level during MIS 5a."

In the case of MIS 3, the original manuscript adopted a bound on local RSL of -3 m to +1 m, and as the reviewer points out this leads to inconsistencies with some marine indicators. To rectify this, we adopt a narrower range that is consistent with the entire dataset. In particular, we modify our lower bound on RSL to correspond to the highest elevation marine indicator at -0.9 m. We also clarify the selection of the upper bound and its relationship with terrestrial markers. The revised text reads (line 78):

“We thus adopt the elevation of the highest of these marine indicators, -0.9 m as the lower bound. Regarding the upper bound, three terrestrial indicators, with ages between 50 and 35 ka, show a consistent constraint on the sea-level highstand of 1 m. Two terrestrial indicators dated to earlier in this time window (Mallinson et al., 2008) are found at lower elevations, however these may represent deposition during a time of rising sea-level rather than during the peak sea-level highstand.”

In addition we add discussion of other terrestrial markers (TCK-19) during the MIS 3 range in the Supplementary Material (lines 14-18).

2. In this study, RSL predictions are generated for a single location, indicated by the white star in Figure 1. In reality, RSL change will not have been the same over the whole area covered by the field data (please add a scale to the inset map in Figure 1). Therefore, in the same way that you explore the sensitivity of the model output to ice history and earth rheology, please also provide brief justification that the RSL predictions generated for the location of the white star are representative of RSL change across the whole study area. In particular, please confirm that the predictions sufficiently represent RSL change at the location of the key sea-level indicators (those used to define the RSL range for MIS 5a and MIS 3).

RSL predictions do vary geographically over the region shown in Figure 1, however RSL high stand predictions at each location marked in the map inset of Figure 1 are within ± 2 m of the prediction at the site shown by the white star (Figure 1). In fact, the “representative” site denoted by the white star is located between the specific sea-level indicators used in defining the RSL range at MIS 3, and the geographical variability amongst this reduced set of sites much smaller (~ 0.5 m). We have edited the following text (lines 107-113) to clarify this point:

“We report RSL predictions at a representative site within the Albemarle Embayment (white star on inset of Figure 1) for a representative time (44 ka) within the middle of MIS 3 (50-35 ka). This representative site lies within the latitudinal range of the reported geological sea-level markers used to define the bound on local peak MIS 3 sea level (Figure 1). We have found that RSL highstand predictions for this reference site differ from field locations by less than 0.5 m.”

We also add a scale to the inset map in Figure 1.

3. Why do you explore the sensitivity to lower mantle viscosity but not upper mantle viscosity? I note that a recent study by Love et al. [2016], which uses US East Coast RSL data to infer regional mantle viscosity values, determines an optimal upper mantle viscosity value for this region that is greater than the value used in this study. The sensitivity of the RSL predictions to upper mantle viscosity should be briefly explored, in the same way that it is for lower mantle viscosity.

Love, R., G. A. Milne, L. Tarasov, S. E. Engelhart, M. P. Hijma, K. Letychev, B. P. Horton, and T. E. Tornqvist (2016), The contribution of glacial isostatic adjustment to projections of sea-level change along the Atlantic and Gulf coasts of North America, *Earth's Future*, 4(10), 440-464, doi:10.1002/2016ef000363.

We agree that it would be useful to explore the sensitivity of our predictions to the choice of upper mantle viscosity. One of the most important (and perhaps the most robust) inferences of mantle viscosity beneath the US east coast within the GIA literature is the seminal study of Potter and Lambeck (*EPSL*, 2003), who demonstrated that a strong gradient in observed MIS5a and 5c high stands extending from Virginia to Barbados could be reconciled by a GIA model with an upper mantle viscosity of 5×10^{20} Pa s. Our group has recently been revisiting this data set (Creveling et al., in

press). The figure below, appearing in that study, shows the geographic variability in the average upper mantle viscosity inferred from seismic shear wave tomography model S40RTS (Ritsema et al., 2011) following the method described in Austermann et al. (2013). The contours show the logarithm of viscosity relative to the site shown by the white dot in the center of the continent. Based on this inference of upper mantle viscosity we expect values to be lower than average (in North America) for the U.S. mid-Atlantic region. On this basis, we believe that the high upper-mantle viscosity preferred by Love et al. (2016), 3×10^{21} Pa s, is unlikely to be robust.

To explore this issue more directly, we repeated our sea level calculations using the viscosity profile preferred by Love et al. (2016), characterized by a lithospheric thickness of 72 km, and an upper and lower mantle viscosities of 3×10^{21} Pa s and 30×10^{21} Pa s, respectively, and the ice history ICE_{PC2}. The figure below, analogous to Figure 5 in the main text, shows a map of the predicted sea level high stand along the U.S. mid-Atlantic region during our MIS 3 time window. Terrestrial sea-level indicators are labeled by elevation in the Chesapeake Bay; these require that relative sea level remained below -7.5 m at mid-MIS 3. However, the simulation based on the Earth model preferred by Love et al. (2016) predicts that the sea level high stand reached ~5 m, i.e., ~12 m higher than the observational constraint.

Nevertheless, to respond to the reviewer's suggestion, we have performed a more thorough sensitivity analysis in which we vary one at a time the upper mantle viscosity from $3\text{-}10 \times 10^{20}$ Pa s, the lower mantle viscosity from $5\text{-}30 \times 10^{20}$ Pa s, and lithospheric thickness from 72-127 km (a value of 96 km was used in the original analysis) from the values that defined the standard model used in the main text. These results are summarized on the new Supplementary Figure S5 shown here:

To incorporate these new results, we have made two changes to the text. First, we added the following sentence on line 250 of the main text:

“We ran simulations with an additional six Earth models to assess the sensitivity of RSL predictions to lithospheric thickness and upper mantle viscosity (see Supplementary Material and Figure S5).”

Second, we added the following text to the Supplementary Material on line 173:

“To explore the sensitivity of our results to other Earth model parameters, we ran simulations using a suite of Earth models where we varied values of the lithospheric thickness and upper and lower mantle viscosity. The resulting RSL predictions for MIS 3, based on the ice model ICE_{PC2} (Figure 2A), for the representative site in Albemarle Embayment (white star, Figure 1) are plotted in Figure S5. In the figure, results are shown for predictions in which the following Earth model parameters were adopted: (A) lithospheric thickness of 72 km (dark blue), 96 km (white) and 127 km (pink); (B) upper mantle viscosity of 0.3×10^{21} Pa s (dark blue), 0.5×10^{21} Pa s (white), and 1×10^{21} Pa s (pink); and (C) lower mantle viscosity of 5×10^{21} Pa s (dark blue), 15×10^{21} Pa s (white) and 30×10^{21} Pa s (pink).”

4. Related to the above point: you discuss (on line 208) the trade-off between a reduction in the direct gravitational effect (which results in sea-level fall in the scenario considered here) and a reduction in the crustal deformation component (which results in sea-level rise). Will the magnitude of the crustal deformation component depend on upper mantle viscosity, and could the adoption of a different upper mantle viscosity lead to a scenario in which the crustal deformation effect does not dominate (see line 208)? The reduction in the magnitude of the peripheral bulge seems to be an important factor that enables you to fit the sea-level data (dashed blue line, Figure 2B), so uncertainty on the magnitude of this effect (due to uncertainty in lithosphere thickness or upper mantle viscosity) should be explored.

Varying the Earth structure will, of course, have no impact on the direct gravitational effect of the ice load. However, since the ocean redistribution is a function of the Earth model, the direct gravitational effect of this component of the surface load will be a function of the Earth model. Moreover, as the reviewer points out, the crustal deformation will be sensitive to variations in the adopted Earth structure. Thus, changing this structure may lead to different relative contributions to sea-level change from deformation and the direct gravitational of the loading. The analysis summarized in the original Supplementary Figure S1, and cited at line 208 in the original manuscript, was focused on identifying the specific reason that ice model ICE_{PC2} improved the fit to the MIS 3 data relative to ICE_{PC} rather than addressing any Earth model dependence in the result. Nevertheless, to address the reviewer's comment we have repeated the analysis of Figure S1 and computed the contribution of deformation

and the direct gravitational effect on RSL predictions at 44 ka for a suite of models in which Earth structure is varied as discussed in the last response and the ice model ICE_{PC2} (Figure 2A) is adopted. The result is plotted in the new Figure S2 shown here:

Two revisions have been made to the text in regard to these new results. First, a sentence has been added to the paragraph mentioned by the reviewer to read (line 221):

“We explore the sensitivity of the RSL decomposition to variations in Earth structure in Supplementary Material, where we adopt the ICE_{PC2} ice history and consider a range of lithospheric thicknesses, and upper and lower mantle viscosities (see Figure S2).”

Second, the following text has been added to the Supplement (line 67):

“The relative contribution of crustal deformation to predicted RSL will be a function of the Earth model, and in particular the values adopted for lithospheric thickness and lower and upper mantle viscosities. Using the Earth model described in the main text, we noted that crustal deformation dominates the RSL signal due to direct gravitational effects. We explore the sensitivity of this decomposition to variations in Earth structure by running simulations for additional Earth models. In Figure S2, results are shown for predictions in which the following parameter ranges are considered: (A) lithospheric thickness from 72 - 127 km; (B) lower mantle viscosity from [5-30] x 10²¹ Pa s; and (C) upper mantle viscosity from [0.3-1] x 10²¹ Pa s. We note that several of these Earth models predict local RSL values that lie outside the observational bound at MIS 3 (Figure S2, orange box). For predictions that are consistent with this bound, the crustal deformation signal is larger than the signal from direct gravitational effects of the surface mass load”

5. The idea of using global ice volume to define the ice extent at a given time (lines 114-115) is a neat way of getting around the problem of working out how ice was spatially distributed during the build-up to the LGM. However, as demonstrated by your results relating to ice extent in the eastern Laurentide, this assumption will not always have held in reality. Given that the partitioning of ice mass between the different ice sheets has a fundamental influence on the pattern of RSL at a given time; would your results still hold if the distribution of ice outside the Laurentide Ice Sheet (LIS) was very different to that assumed here? Addressing this issue in a robust manner is outside the scope of this study, but a brief discussion of whether it could negate the results is warranted.

RSL predictions at the Albemarle Embayment will be relatively insensitive to changes in the distribution of ice outside of the LIS because the embayment is in the far field of the other ice sheets. The signal from these ice sheets will primarily be related to deformation associated with the total ocean load and rotational effects from the mass redistribution. These effects are not sensitive to the details of ice geometry in the far field of the embayment, and they will be small compared to near-field effects of gravitational attraction and crustal deformation associated with the LIS. As an example, we performed a test run in which we distributed ice uniformly over Fennoscandia and Antarctica (rather than the western LIS) and found that the predicted RSL was perturbed by only ~0.6 m. In light of these results we have added the following note about the sensitivity of the RSL predictions to the distribution of ice outside LIS in the Supplementary Material (lines 102-108).

“We note that while RSL predictions at the Albemarle Embayment are sensitive to ice geometry within the eastern sector of the LIS, these predictions will be relatively insensitive to the geographic distribution of ice cover in regions in the far-field of the mid-Atlantic U.S. East Coast. As an example, we performed a simple test in which the ice removed from the eastern sector of the LIS to construct the ICE_{PC2} model was distributed uniformly over Fennoscandia and Antarctica (rather than over the western sector of the LIS) and found that the predicted sea level high stand at our test site at 44 ka was only perturbed by 0.6 m.”

6. Changing the geometry of ice in the eastern sector of the LIS seems to have a significant effect on RSL predictions at the study site (e.g. lines 228-231) so I took some time to try to understand how the various ice history scenarios differed. However, I found it difficult to follow the description of the different ice sheet geometries (lines 88-99 in the supplementary information), partly because the text did not always seem to agree with the information shown in Supplementary Figure 2. For example, panel B of this figure shows configuration PC2, in which the latitudinal boundary for no-ice is apparently 60° N (line 91 of supplementary material). But in the figure, the southern margin of the ice sheet does not just terminate at 60° N across eastern Laurentide; do you also invoke some (sensible) assumption relating to the fact that ice extent will also be governed by the underlying topography? What are the implications of this for PC3 and PC4 (which are not shown, but apparently have a different latitudinal extent)? I was also confused by the statement that Baffin Island is not glaciated in Geometry 2 and Geometry 3 (lines 94-95), since it does appear to be partly glaciated in Supplementary Figure 2D. Does the statement just relate to the southern part of the island? There is also reference to Figure 4B in the caption to Supplementary Figure 2, which does not seem to exist.

Thank you for pointing out the inconsistencies within the text. First, we have corrected the mistake in the caption of Figure S3 (Figure S2 in original manuscript). In addition, we have corrected the text describing the glaciation of Baffin Island (line 125) by noting that only the southern portion of Baffin Island is deglaciated in ice models Geometries 2 and 3. We have also modified Figure S3 to better reflect the ice models described in the text and adopted in the numerical calculations. There is no assumption made about the underlying topography; rather the ice cut-off is at 60° N, 57° N, and 55° N, respectively, for the PC2, 3, and 4 versions of the ice history.

Minor comments

1. Title: Should ‘Sea-Level Records’ be hyphenated (you seem to follow this rule in all other cases where ‘sea level’ is used as a compound adjective)?

We agree. This has been changed accordingly.

2. Line 13-14: ‘paleo sea-level values near present-day’ reads a little odd as ‘present-day’ refers to a time; perhaps rephrase to clarify that you are referring to the fact that MIS 3 sea level indicators are found close to the elevation of the current shoreline.

This sentence has been rephrased to read: “at the present-day shoreline of”

3. Lines 52-53: the text on these lines is a little ambiguous: it is not clear whether you are suggesting that previous GIA calculations predict that the land where the shoreline records are found was 70 m below sea level during MIS 3, or whether previous GIA calculations predict that the MIS3 shoreline records should currently be 70 m below sea level, rather than close to present sea level. (I think the latter is correct)

We have clarified this sentence by revising the phrase “should be found” to “should presently be found” (line 52).

4. Please briefly mention whether sediment compaction is likely to have affected any of the sea-level indicators.

We have revised the text to cite a study analyzing compaction rates over the Holocene by adding the phrase “and sediment compaction has led to only minor subsidence in this region (Brain et al. 2015)”

to the end of the sentence (line 55).

5. Lines 13-114: ‘...we make the standard assumption...’ – is this just an assumption that this adopted throughout this study, or is it an assumption that has been used in previous studies? If the latter, please include a reference.

This assumption has been made in previous studies, and we include, as an example, a citation to Raymo et al. (2011).

6. Line 117: it is not quite clear what ‘this combination’ refers to, because the train of logic is slightly interrupted by the mention of exploring other model combinations in the previous sentence.

We have clarified this issue by now specifically citing the ICE-5G model (line 127): “Using the combination of the ICE-5G model and Earth structure described above”.

7. Lines 182-184: In the description of the scenarios where the eastern sector of the LIS is ice free you mention that ice is re-distributed over the western LIS and the Fennoscandian Ice Sheet (FIS). Please give an indication of what this means in terms of ice thickness added to the western LIS and the FIS, so that readers can determine whether this is a reasonable modification to the existing ice history in these regions.

We agree that this information will be useful for the reader. To clarify, we have updated the text to read (line 193):

“the ice removed in this exercise, equivalent to 6.8 m of GMSL, is distributed uniformly over the western sector of the LIS, and the Cordilleran and Fennoscandian Ice Sheets. The latter resulted in an increase in ice thickness of ~170 m in each region.”

In addition this information is also added to the Supplementary Material (line 99).

8. Line 219: the ranges quoted seem to be incorrect (correct values are given in the supplementary material)

Thank you for pointing this error out. The values in the main text have been updated to match those in the Supplementary Material.

9. Figure 3B: It is difficult to distinguish the blue lines from the purple lines

To address this issue, we have changed the purple lines to black, and made them semi-transparent so that it is easier to compare the predictions based on the standard ice geometry to those that adopted an ice-free eastern sector of Laurentia.

10. Since you use a continuous colour scale in Figure 5, please label the contours to make it easier to interpret this figure

Contour labels have been added to the plot.

11. Supplementary Figure 1: please clarify in the caption to this figure that predictions are for the location of the white star in Figure 1

We have now included this information in the caption

Response to Reviewer #2 :

This manuscript examines some previously enigmatic MIS 3 sea level constraints along the U.S. East Coast from a GIA modeling perspective. They show that the fairly “high” elevations of the relative sea level markers can be explained using a revised geographic extent of ice cover in the Laurentide ice sheet during this time interval. This paper is a valuable contribution and is appropriate for the audience

of Nature Communications both in terms of the value of this particular study in explaining these MIS 3 data (that I have personally wondered about) and also underscores the ever important message to the broader community that local, or relative sea level can indeed be very different than global mean sea level, which remains an extremely valuable message for the broader community to appreciate.

My comments are minor, and I would anticipate that this manuscript could be accepted after some of the minor revisions below are addressed:

Line 35: I think you mean “reconstructions” rather than “complications”

Actually, our intent here was to point out that the mapping from oxygen isotopic values to global ice volumes faces a series of complications that lead to discordant estimates of global mean sea level during MIS 3. We have clarified the text by revising the phrase to read (line 34):

“uncertainties in this correction and other complications in mapping isotope values to ice volumes have yielded estimates of peak MIS 3 GMSL”

Line 38: instead of “ice cover” I think it is clearer to say “former ice sheets” or “former ice sheet extent”

We have changed the wording to “former ice sheets” (line 37).

Line 41: consider re-phrasing to “present-day elevation”

Done.

Line 61: spelling of estuarine

This has been corrected.

Line 96: theory of what?

We have clarified this phrasing to read “sea-level theory” (line 103).

Line 100-102: What is the effect of this assumption? I.e., how valid is it to assume that sea level is the same at the site of the white star and for the remaining sites where the data come from? Is there significant regional heterogeneity in the RSL signal?

This question was also raised by Reviewer #1. As we noted above, RSL predictions do vary geographically over the region shown in Figure 1, however RSL high stand predictions at each location marked in the map inset of Figure 1 are within ± 2 m of the prediction at the site shown by the white star (Figure 1). In fact, the “representative” site denoted by the white star is located between the specific sea-level indicators used in defining the RSL range at MIS 3, and the geographical variability amongst this reduced set of sites is much smaller (~ 0.5 m). We have edited the following text (lines 107-113) to clarify this point:

“We report RSL predictions at a representative site within the Albemarle Embayment (white star on inset of Figure 1) for a representative time (44 ka) within the middle of MIS 3 (50-35 ka). This representative site lies within the latitudinal range of the reported geological sea-level markers used to define the bound on local peak MIS 3 sea level (Figure 1). We have found that RSL highstand predictions for this reference site differ from field locations by less than 0.5 m.”

Figure 2. The subscripts of the different model names is near impossible to read with the naked eye. The black dotted line seems a lot thicker in the legend for some reason.

The text associated with model names has been enlarged to improve readability and the thickness of the dotted lines in the legend has been changed to make them consistent with the plot.

Line 137: what is “this ice history” ICE-5G? If yes, then explicitly say so.

We now cite ICE-5G on line 147.

Figure 3. The colors (in particular the purple and especially when overlaid by the orange) are very hard to discern here. Perhaps making the purple lines black or gray would increase the ability to distinguish this group of curves visually.

As suggested, we have changed the purple lines to black, and we have also made them semi-transparent so that it is easier to compare the predictions based on the standard ice geometry to those that adopted an ice-free eastern sector of Laurentia.

Figure 4. It is not apparent from the one dotted line, which side of the dotted line the ice is sitting on (though the text explains it). Perhaps a white transparent overlay for the area of the MIS 5a to 3 ice sheet would help here. Perhaps also saying that the dotted line shows the eastern extent instead of extent, would also help?

We have revised the first sentence of the caption to Figure 4 to read: "Geographic map showing ... the eastern extent of the ice model ICE_{PC2} from MIS 5a to MIS 3 (dotted black line)"

Line 252: insert the word "that" after "show"

Done.

Supplementary Fig. 2: you reference figure 4B multiple times here, but there is no panel B in Fig. 4 of the manuscript. Upon examining this figure, I am now very confused by figure 4 in the text. There seems to be a discrepancy here. In caption for this supplementary figure you need to say what time frame is represented in these diagrams... i.e., geography distribution of ice WHEN?

Thank you for catching this error. We have updated the caption in Supplementary Figure S3 (Supplementary Figure S2 in original manuscript) to refer to the correct figure, Figure 3B.

Response to Reviewer #3:

This manuscript presents a new comparison of existing sea level data from the U.S. mid-Atlantic with an ensemble of Glacial Isostatic Adjustment (GIA) model simulations for MIS3. Recently published data (Pico et al., 2016, QSR) suggest that global mean sea level was higher than previous estimates, implicating a reduced global ice volume during this time. While considering a variety of parameters in the GIA, the authors indicate that US Mid-Atlantic sea level records for MIS3 can only be explained with a reduced ice volume over the eastern sector of the Laurentide Ice Sheet (LIS). The authors appear to have done their due diligence in assessing the parametric uncertainty in their GIA model (mantle viscosity, MIS5a&MIS5c sea level uncertainty, eastern LIS volume reconstruction uncertainty, etc). The findings because most pre-LGM ice extent and volume constraints are poor due to the destruction of potential records by the LIS at maximum extent. This work will provide useful constraints on ice sheet boundary conditions for the modelling community. The manuscript is well written and provides a clear description of the mechanisms controlling U.S. mid-Atlantic sea level change during MIS3. I support this manuscript for publication, but I have a few questions and minor comments that should be considered to clarify some of the methods used in this analysis:

Line 74: It is not completely clear to me why the authors select 1m as the upper bound for their MIS3 sea level range. The authors use Qtp-1 as their upper terrestrial limiting sample, but there appear to be other terrestrial records from Mallinson et al. (2008) that are a bit lower. It does not appear it will change the conclusions, but I would like a better description as to why Qtp-1 was selected for the upper bound.

This concern was also raised by Reviewer #1. In the case of MIS 3, the original manuscript adopted a bound on local RSL of -3 m to +1 m, and this lead to inconsistencies with some marine indicators. To rectify this, we adopt a narrower range that is consistent with the entire dataset. In particular, we modify our lower bound on RSL to correspond to the highest elevation marine

indicator at -0.9 m. We also clarify the selection of the upper bound and its relationship with terrestrial markers. The revised text reads (line 79):

“Regarding the upper bound, three terrestrial indicators, with ages between 50 and 35 ka, show a consistent constraint on the sea-level highstand of 1 m. Two terrestrial indicators dated to earlier in this time window (Mallinson et al., 2008) are found at lower elevations, however these may represent deposition during a time of rising sea-level rather than during the peak sea-level highstand.”

We have also included the following text in the Supplementary Material on lines 16-18:

“These two samples represent the minimum elevation of terrestrial sea-level indicators whose age uncertainties span 50-35 ka.”

We now refer to the sample TCK-19 in Mallinson et al. (2008) on line 14 (Supplementary Material) and explain that the lower elevation markers do not capture the span of the MIS 3 interval 50-35 ka, and therefore may not represent the MIS 3 sea-level highstand.

Line 140-141: Please report the full range of GMSL values for MIS 5a and 5c from Creveling et al and why you select -15m and -10m, respectively, as the optimal values for your initial model runs. This gets discussed later in the sensitivity tests (lines 211-223), but it would be good to introduce the full range here.

Agreed. The full range of GMSL values preferred on the basis of the analysis in Creveling et al. is cited in the following text (line 152):

“Second, we adopt GMSL values of -15 m and -10 m for MIS 5a and 5c, respectively; these values are within bounds (5a: -18 m to 0 m, 5c: -20 m to 1 m) derived by Creveling et al.”

Figure 2b and 3b: Do the MIS 5a and MIS 3 sea level boxes include the +/- 3m paleotidal uncertainty. If so, perhaps consider mentioning this in the figure description so there is not confusion as to why vertical range is larger than in figure 1.

These ranges do include elevation uncertainty. We have revised the captions of Figure 2 and 3 to add the phrase: “including the ± 3 m paleotidal uncertainty”.

Figure 3b: Would it be possible to increase the transparency of the blue ‘standard ice geometry lines.’ As is, it is impossible to see the bottom of the range of the purple lines (‘ice-free Eastern Laurentide’).

This suggestion has been adopted in revising the figure.

Line 215-216: “-11 m to -1 m”: check the negative sign in front of -1. It looks like it is a dash.

This has been corrected.

Line 219: The Creveling et al. ranges, “0-16 m” and “0-20 m”, should be negative, right?

Yes. This has been rectified.

Supplementary Figure 1: flip x-axis so that it follows the same direction as the figures of the main text.

As suggested, we have flipped the axis.

Supplemental Line 78: Please provide further description as to how you distribute the eastern sector LIS volume over western LIS and Fennoscandia. Is this equally distributed? Are there any other RSL records that could constrain where you put this ice? Why not put this ice in Antarctica?

The excess ice is uniformly distributed over the LGM extent of the western LIS, the Cordillera and Fennoscandia. We have revised the text to read (line 193):

“the ice removed in this exercise, equivalent to 6.8 m of GMSL, is distributed uniformly over the western sector of the LIS, and the Cordilleran and Fennoscandian Ice Sheets. The latter resulted in an increase in ice thickness of ~170 m in each region.”

We did not put ice in the Antarctic because recent studies have tended to suggest that the volume of the Antarctic Ice Sheet at LGM may have been significantly smaller than many previous global ice models have suggested (Whitehouse et al., *QSR*, 2012, Ivins et al., *GJR*, 2013).

REVIEWERS' COMMENTS:

Reviewer #1 (Remarks to the Author):

Second review of 'Sea-Level Records from the U.S. Mid-Atlantic Constrain Laurentide Ice Sheet Extent During Marine Isotope Stage 3' by Pico et al.

The authors have done an excellent and thorough job of addressing the comments raised in my original review. After reading the revised manuscript I have a small number of very minor comments, but in general am very happy to recommend this article for publication.

Minor comments

1. Caption to Figure 1: the timing of MIS 5e is not included in the list of ages.
2. In a couple of places references to purple lines in Figure 3 remain (these lines are now black)
3. Line 219: you talk about 'reduced crustal deformation'; the initial conclusion I drew from this phrase was that there was less downwards deflection of the earth surface (due to the smaller ice load), but this did not tie in with your statement that this results in sea-level rise. In fact, I think the site is far enough from the ice sheet that the effect you are talking about is less upwards deflection of the earth surface (smaller peripheral bulge). It would be useful for the reader if you could remind them of the direction of the earth deformation at this location.
4. Line 249: suggest 'We also ran...'
5. I think that the infill of the triangular symbols in Figure 5 uses the same colour scale as the background plot – if this is true then perhaps mention this in the caption.
6. Supplementary Information, line 70: the text 'crustal deformation dominates the RSL signal due to direct gravitational effects' is a little confusing. I think you are simply saying that the crustal deformation signal is greater than the signal due to gravitational effects?
7. Supplementary Information, line 100: 'an increase in ice thickness...?'
8. Supplementary Information, line 144: 'the Cordilleran Ice Sheet'?

Pippa Whitehouse

Reviewer #3 (Remarks to the Author):

The revised manuscript has been adequately updated to address the concerns in my initial review. This study provides a highly novel approach toward rectifying Laurentide Ice Sheet volume during MIS3 by comparing paleo sea-level indicators from eastern North America with the results from a glacial isostatic adjustment model. I appreciate the detailed assessment of parametric uncertainty in the GIA model, and I find the results to be convincing, with appropriate caveats when necessary. Since very few studies have attempted to quantify Laurentide ice volume during MIS3, these results will be highly valuable to researchers interested in paleoclimate and glacial geology. In particular, these results will help contribute to the ice sheet modeling community who may be looking for data-constrained ice sheet boundary conditions leading into the Last Glacial Maximum.

Finally, this manuscript is well-written and provides a good summary of the factors controlling sea-level change during MIS3. I have enjoyed reading both the original and revised editions.

Response to the Reviews

We thank the three referees for the second round of positive and constructive reviews of our original manuscript. Our appreciation is reflected in the acknowledgements. As we detail below, we have revised the manuscript to address each of the comments raised by the reviewers. In the following, we intersperse the reviewers' comments (black font) with our responses (in blue font).

Response to Reviewer #1:

Second review of 'Sea-Level Records from the U.S. Mid-Atlantic Constrain Laurentide Ice Sheet Extent During Marine Isotope Stage 3' by Pico et al.

The authors have done an excellent and thorough job of addressing the comments raised in my original review. After reading the revised manuscript I have a small number of very minor comments, but in general am very happy to recommend this article for publication.

Minor comments

1. Caption to Figure 1: the timing of MIS 5e is not included in the list of ages.

We now include the timing of MIS 5e (labeled at 120 ka).

2. In a couple of places references to purple lines in Figure 3 remain (these lines are now black)

These references have been corrected.

3. Line 219: you talk about 'reduced crustal deformation'; the initial conclusion I drew from this phrase was that there was less downwards deflection of the earth surface (due to the smaller ice load), but this did not tie in with your statement that this results in sea-level rise. In fact, I think the site is far enough from the ice sheet that the effect you are talking about is less upwards deflection of the earth surface (smaller peripheral bulge). It would be useful for the reader if you could remind them of the direction of the earth deformation at this location.

We have added the phrase: "a smaller upward deflection of the Earth's surface expressed as sea-level rise." (line 171)

4. Line 249: suggest 'We also ran...'

We have replaced "We ran..." with "We also ran..." according to this suggestion.

5. I think that the infill of the triangular symbols in Figure 5 uses the same colour scale as the background plot – if this is true then perhaps mention this in the caption.

We have edited the caption to include information about the colorbar and the location of the triangles. It now reads:

Elevation of sea level data are plotted as triangles at the Chesapeake Bay (-7.6 m, -1.7 m, -3.4 m) and the Hudson shelf (-30 m), where the color represents the elevation shown by the colorbar. Upper bounds on sea level are represented by downward pointing triangles, while lower bounds are plotted as upward pointing triangles.

6. Supplementary Information, line 70: the text 'crustal deformation dominates the RSL signal due to direct gravitational effects' is a little confusing. I think you are simply saying that the crustal deformation signal is greater than the signal due to gravitational effects?

Yes. We have updated this sentence to read "crustal deformation dominates the RSL signal compared to direct gravitational effects".

7. Supplementary Information, line 100: 'an increase in ice thickness...?'

Yes, we have updated this sentence to read "an increase in ice thickness..."

8. Supplementary Information, line 144: 'the Cordilleran Ice Sheet'?

We have updated the text to read "the Cordilleran Ice Sheet" according to this suggestion.

Response to Reviewer #3:

The revised manuscript has been adequately updated to address the concerns in my initial review. This study provides a highly novel approach toward rectifying Laurentide Ice Sheet volume during MIS3 by comparing paleo sea-level indicators from eastern North America with the results from a glacial isostatic adjustment model. I appreciate the detailed assessment of parametric uncertainty in the GIA model, and I find the results to be convincing, with appropriate caveats when necessary. Since very few studies have attempted to quantify Laurentide ice volume during MIS3, these results will be highly valuable to researchers interested in paleoclimate and glacial geology. In particular, these results will help contribute to the ice sheet modeling community who may be looking for data-constrained ice sheet boundary conditions leading into the Last Glacial Maximum.

Finally, this manuscript is well-written and provides a good summary of the factors controlling sea-level change during MIS3. I have enjoyed reading both the original and revised editions.

We appreciate these positive comments and thank the reviewer for the previous constructive comments we received on the earlier version of this manuscript.